

# Exploring the Applicability and Limitations of Selected Optical Scattering Instruments for PM Mass Measurement

Jie Zhang[*], Joseph P. Marto, James J. Schwab

Atmospheric Sciences Research Center, University at Albany, State University of New York, NY, USA

*Correspondence to: Jie Zhang, Email: jzhang35@albany.edu

**Abstract:** Two optical scattering instruments for particle mass measurement - the Thermo Personal Data RAM (PDR-1500), the TSI Environmental DustTrak DRX (Model 8543) were evaluated by 1) using poly- and mono-disperse test aerosol in the laboratory, and 2) sampling ambient aerosol. The responses of these optical scattering instruments to different particle characteristics (size, composition, concentration) were compared with responses from reference instruments. A Mie scattering calculation was used to explain the dependence of the optical instruments' response to aerosol size and composition. Concurrently, the detection efficiency of one Alphasense Optical Particle Counter (OPC-N2) was evaluated in the laboratory as well. For ambient aerosol, a moderate coefficient of determination ($R^2$=0.64) with positive slope was found between aerosol mass median diameter and the ratio of PDR reported mass to that measured by an Aerosol Mass Spectrometer (AMS). Aerosol size was verified to be the primary factor affecting optical response in this study, but aerosol chemical composition and refractive index were also evaluated for their influence. These observations and calculations help evaluate the applicability and limitations of these optical scattering instruments, and provide guidance to designing suitable applications for each instrument by considering aerosol sources and aerosol size.

## 1 Introduction

The measured mass concentration of fine particulate matter (PM2.5) both indoors and outdoors is important for many reasons (Chow et al., 2005; McMurry, 2000; Brauer et al. 2011). A major reason is that fine particulate matter is associated with adverse effects on human health, specifically increased morbidity and mortality rates (Dockery et al., 1993; Landen et al., 2011). The standard mass concentration measurement method established by the U.S. Environmental Protection Agency (EPA) for compliance with National



Ambient Air Quality Standards (NAAQS) is based on gravimetric filter sampling and weighing (Sousan et al., 2016a), which are repeatable and have well-characterized accuracy and precision. However, these methods cannot provide real-time aerosol mass concentration, which limits the information available regarding aerosol sources, diurnal variation (Wallace et al., 2011), and high concentration spikes of short duration (Chung et al.,

5    2001).

These drawbacks can be avoided by using real-time continuous instruments, most commonly the Tapered Element Oscillating Microbalances (TEOM) or Beta Attenuation Monitors (BAM) (EPA, 2013). However, due to the high cost and large size of these instruments, it is difficult to deploy these real-time instruments for quick-response situations or in a wide spatial coverage, especially in remote areas and developing countries.

Less expensive, portable, small sensors which use light scattering to infer particle mass concentration have become available in the past few decades, and they are currently drawing much attention as an alternative to the well-established methods described above (Hinds et al., 1999; Holstius et al., 2014; Wang et al., 2015; Wang et al., 2016).

Typically, these light scattering sensors are calibrated by the manufacturer using a specific test aerosol, which

may or may not be representative of ambient testing conditions at a given location. For this reason, there is ongoing interest in evaluating the capabilities and limitations of optical scattering sensors in the laboratory when challenged with aerosol of varying sizes and compositions. Sousan et al. (2016b) found the OPC-N2 (Alphasense Ltd) performed similarly to the PAS-1.108 (Portable Aerosol Spectrometer, GRIMM 2010) for particles with diameter above 1 μm. In other studies, laboratory measurements were complemented by field

comparisons with reference instruments (such as TEOM or gravimetric filter methods). Wallace et al. (2011) suggested a calibration factor of 0.38 for the DustTrak when sampling ambient aerosol after comparing the instrument with TEOM data.

Investigating the fundamental performance of these optical instruments for different kinds of aerosol aids in understanding their properties and guiding their suitable use. Based on these considerations, experiments were

designed and performed to evaluate three distinct optical sensors - the Thermo Personal DataRAM (PDR-1500), the TSI Environmental DustTrak DRX (Model 8543), and the Alphasense Optical Counter

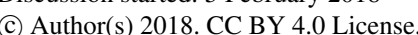



(OPC-N2) - using mono-disperse aerosol to study the instrument's dependence on particle size, and poly-disperse test aerosol to study the effect of aerosol refractive index (related to chemical composition). A Mie scattering calculation was used to quantitatively describe the performance of these instruments. Optical instruments readings were then related to the aerosol size measured by Scanning Mobility Particle Sizer

(SMPS) and chemical composition of ambient aerosol measured by a High Resolution Time-of-Flight Aerosol Mass Spectrometer (HR-TOF-AMS) to provide detailed analysis of the effects of ambient size and chemical composition on the response of these optical instruments. Using these laboratory results, the average refractive index of the ambient aerosol was estimated based on the response of optical instruments and the aerosol size distribution.

## 2 Methods

### 2.1 Instrumentation

The TSI DustTrak DRX (model 8543, hereafter DRX) is a combined photometer and optical counter, which uses a 655 nm laser to illuminate a sample stream and measures the intensity of scattered light perpendicular to the stream with a photodetector (Wang et al., 2009; TSI Inc., 2017). The measured intensity is a function of

the total particle volume, the particle refractive index, and the particle shape (Wallace et al., 2011). Scattering intensities of individual particles are used to group particles into broad size bins - typically $PM_1$, $PM_{2.5}$, $PM_4$, and $PM_{10}$. The DRX is calibrated by TSI using A1 dust (Arizona road dust). The calibration factor was set to the default value of 1 for this study.

The PDR-1500 (Thermo Fisher Scientific Inc., 2014, herafter PDR) is a highly sensitive nephelometric

monitor using an 880 nm wavelength LED source for particle scattering measurement. The PDR-1500 was also calibrated by the manufacturer using Arizona road dust (Sullivan et al., 2014).

The OPC-N2 (Alphasense Ltd., hereafter OPC-N2) measures the intensity of light scattered from particles using an optical receiver at a forward angle of approximately 30° (Sousan et al., 2016b). The light intensities are binned and reported as particle counts in 16 size bins from 380 nm to 170 μm. The laser wavelength of

OPC-N2 is 658 nm, and instrument specifications list a lower detection limit for particle size of 380 nm. The



number and sizing of the OPC-N2 were calibrated by the manufacturer using polystyrene latex (PSL) particles, and a constant density of 1.65 g cm$^{-3}$ is used for converting the number concentration to mass concentration.

Our reference instruments included a TSI Scanning Mobility Particle Sizer (SMPS), which consisted of an Electrostatic Classifier (EC, model 3080), a Differential Mobility Analyzer (DMA, model 3081), and a

Condensation Particle Counter (CPC, model 3785). The SMPS operated with a 10:1 sheath-to-sample flow ratio (sheath flow 3 L min$^{-1}$, and sample flow 0.3 L min$^{-1}$), which led to an effective measured aerosol size range of 14.9 nm to 673.2 nm. A tapered element oscillating microbalance (TEOM) mass monitor (R&P Model 1400ab), provided continuous gravimetric-based non-volatile aerosol mass concentrations with a time resolution of 5 mins (Hogrefe et al., 2004). An Aerodyne HR-TOF-AMS (hereafter AMS) was also used to

measure the chemical composition and composition mass size distribution in real-time for non-refractory sub-micron aerosol (Drewnick et al., 2005; Sun et al., 2012). A collection efficiency (CE) factor of 0.5 was used to account for the aerosol loss caused by aerosol bounce at the vaporizer and aerosol lens transmission (Zhang et al., 2005, Canagaratna et al., 2007, Sun et al., 2009). The default relative IE (RIE) of organic compounds (1.4), nitrate (1.1), sulfate (1.2), ammonium (4), and chloride (1.3) were used (Jimenez et al.,

2003; Lee et al., 2016). The AMS-derived chemical composition mass concentration was used to calculate the aerosol density used by SMPS, as described in the Supplement.

## 2.2 Experimental setup

The experiments were conducted first as a laboratory evaluation, which evaluated the performance of optical sensors under controlled aerosol conditions, and second as an ambient test, which evaluated the performance

of sensors under ambient aerosol conditions, including the effect of aerosol size and composition on sensor readings.

The laboratory setup used for evaluation and calibration of optical instruments has been described by Hogrefe et al. (2004) and is shown in Fig. 1. The laboratory system consisted of an aerosol generation and dilution system, as well as a 500-liter aerosol chamber with sampling ports. The aerosol generation instrument was a

constant output atomizer (TSI Model 3076). The dilution system reduced the aerosol mass concentration

produced by the atomizer and dried the aerosol using dry air flow. Before sampling by the different instruments, the generated aerosol underwent equilibration in the aerosol chamber. The instruments sampled from the middle of the chamber, and about 20 minutes was required for the instruments to stabilize after each modification to the dilution or generation system. For evaluating the dependence of optical sensors

performance on size, different-sized, mono-disperse PSL particles were used. For each size PSL, dilution liquid was incrementally added during measurement to vary concentration. After the PSL experiments, synthetic particles with four different chemical compositions (NaNO$_3$, (NH$_4$)$_2$SO$_4$, sucrose, adipic acid) were generated for testing the performance dependence on particle composition, as light scattering is affected by particle refractive index (Seinfeld and Pandis, 2016). For each material, the solution concentration and/or the

amount of dilution air was varied during experiments to modify the concentration.

During these experiments, the TEOM was used as the reference instrument for NaNO$_3$, (NH$_4$)$_2$SO$_4$, sucrose, and PSL particles because it had the same sampling location as optical instruments. The SMPS was used as the reference for adipic acid, due to adipic acid's high volatility (Mønster et al., 2004; Seinfeld and Pandis, 2016) and observed loss on the 50° C TEOM mass sensor.

For ambient experiments, two optical sensors (DRX and PDR), one HR-TOF-AMS, and one SMPS were connected to a dryer, to keep the RH below 40%, and then connected to outside air through a vent port. The OPC-N2 was not used for these tests due to its size detection limit (380 nm) being larger than the most ambient aerosol of this study, while TEOM is not used due to its high frequency noise in ambient measurement.

**2.3 Mie scattering calculation**

The Mie scattering calculation followed the techniques in the MATLAB version of Mie theory for homogeneous spheres described by Mätzler (Mätzler, 2002). The required input parameters include the complex refractive index, the sphere radius, light wavelength, and the scattering angle (Li et al., 2017). Considering only spherical particles is reasonable in our situation because smaller non-spherical particles

(<1um) are more similar to their spherical phase than larger ones (Smith, 2009). The scattering angles and

light wavelengths used in these three optical instruments are listed in Table 1, and the scattering angle of 90 degrees for the PDR was used for simplicity. Combining these sensor and aerosol parameters, the relative scattering flux per unit aerosol volume (RF_v) for each particle size was then calculated. Knowing the normalized volume distribution (NVD) of the generated aerosol, the integrated relative flux (hereafter RF)

received by the sensor detector was estimated by summing the product of RF_v and NVD over particle size. RF was then used to evaluate the performance of optical instruments.

## 3 Results and discussion

### 3.1 Performance for mono-disperse particles

The dependence of DRX, PDR, and OPC-N2 performance on particle size was studied using five sizes (90±14 nm, 173±9 nm, 304±9 nm, 490±15nm, and 1030±31 nm) of PSL particles (Ted Pella, Inc.). To compare optical sensors with the reference instrument (TEOM), 15 minutes of data collected at the end of each mass concentration plateau were averaged.

The relative responses of the optical sensors (PDR, DRX, and OPC-N2) compared to the TEOM for the

mono-dispersed PSL particles are shown in Table 2. The outputs of these sensors were linearly regressed against particle mass concentration measured by the TEOM to get the ratio of sensor response to TEOM readings, and all results were well-correlated ($R^2$>0.90). The results from the PDR and DRX for 304 nm PSL particles is shown in Fig. S1 as an example. As particle size increased in these experiments, the responses of the PDR and DRX showed a maximum for one specific size PSL (304 nm for DRX, and 490 nm for PDR).

For both responses, these sizes were closest to half of the light source wavelength used by the sensors. For OPC-N2, this maximum was missed due to its detection limit. For PSLs large enough to trigger a response, the OPC-N2 detection efficiency was still lower than 1 for these two sizes – the detection efficiency was 68% for 490 nm and only 48% for 1030 nm.

Mie scattering calculation results were used to describe the observed performance characteristics of the DRX and the PDR. The relative scattering flux of particles for conditions appropriate to the DRX and PDR is shown in Fig. S2 and Table 2. Figure S2a shows that the RF peaks at about 400 nm for DRX and 550 nm for PDR, which matches the above result that PDR showed its highest relative response to a larger PSL (490 nm) than DRX (304 nm), and as particle size increased beyond peak size, the relative scattering flux decreased.

High correlation coefficients were shown for the RF and the optical instruments to TEOM ratio, with $R^2$=0.97 for DRX and 0.81 for PDR as shown in Fig. S2b. This is indicative of the general positive relation between the calculated RF with optical instrument's response to PSL. Here the value for 90 nm PSL is not considered, since both the DRX and PDR showed high response bias for 90nm, which may be caused by the proximity of

90 nm to their detectors' size limits (100n m for DRX and PDR, Thermo Fisher Scientific Inc., 2014; TSI Inc., 2017), or the uncertainty introduced by using a single scattering angle in the calculation.

## 3.2 Performance for poly-disperse particles

### 3.2.1 Mass concentration dependence on particle composition

To focus on the instruments' dependence on particle composition (mainly the effect of refractive index), three

groups of tests with different concentration liquid samples were sampled, as shown in Table S1. The changes in solution concentration produced a shift in aerosol number/mass distribution sampled by the instruments. In each of the three groups, concentrations were selected such that the volume size distributions of the four compounds were very similar (as shown in Fig. S3).

The ratios of those optical instruments readings to those of the reference instruments for different kinds of

aerosol in each group were determined, as shown in Table 3. A strong, linear relationship between the DRX, PDR, and OPC-N2 with the TEOM for 0.45 g L$^{-1}$ sucrose was found, with $R^2$=0.99, as shown in Fig. S4 as an example. For other kinds of particles and concentrations, the results were similar. This illustrates the high linearity and stability of the responses of these optical instruments.

Table 3 shows the variation of ratios related to the aerosol composition in each group, in addition to an

inter-comparison between different groups. For low dilution concentration (Group 1), the DRX showed a





higher response (ratio vs TEOM) for sucrose (slope=0.92), and $(NH_4)_2SO_4$ (slope=0.85), but lower ratios for

$NaNO_3$ and adipic acid (58% and 61%, respectively). Similar performance was also shown for the PDR and

OPC-N2. A lower detection ratio was observed for the OPC-N2, due to its detection limit (380 nm), as it

would fail to detect many particles. Different ratios of optical sensors for different particles are related to their

refractive index, and the aerosol with a larger refractive index results in a higher measurement ratio. More

detailed discussion will be given in the next section, while the refractive index of these materials is shown in

Table 4 (Weast, 1976; Seinfeld and Pandis, 2016). For moderate and high dilution concentrations, similar

behavior was observed. Generally, the DRX and PDR ratios produced higher ratios for particles with larger

refractive indices. It is worth noting that only bright aerosols were considered, meaning that the imaginary

part of the refractive index of the aerosol was zero.

Since the median diameter of the volume size distributions increased from Group 1 to Group 3 as seen in

Table S1, the increase in optical response shown in Table 3 verifies the size dependence illustrated by the PSL.

That is, the response ratios of the optical instruments increase with increasing particle size for the same

particle material in these three groups. Also, with increasing particle size, the difference between DRX and

PDR decreased from about 2.7 times for the smallest median diameter, to 1.7 times for the largest median

diameter. With size distribution shifting to larger sizes, the PDR displayed a larger relative response (to the

TEOM) due to its longer wavelength.

### 3.2.2 Mie scattering calculation for optical instruments

For Mie scattering calculations, all particles were assumed to be spherical, and their refractive index was

independent of wavelength over the range of interest. Refractive indices of the four materials are shown in

Table 4. Some materials have two or three different refractive indices, the values of which are related to their

crystal structure (Eggleton et al., 1991). To address these materials, the relative scattering flux was calculated

by using the maximum and minimum refractive indices, which produced a range of relative flux values for the

material. The maximum and minimum relative fluxes were then averaged to represent a best estimate relative

scattering flux of the material, and the difference between averaged value and minimum value was used as the

error range. Following the above method, the relative Mie scattering flux of these four particle compounds



was obtained for the optical sensors at different dilution concentrations as shown in Table S2. Here the Mie relative flux was only calculated for the DRX and PDR, and not the OPC-N2. This is because the DRX and PDR responses are directly related the scattering flux, while the OPC-N2 response is directly related to the number of pulses.

The highly correlated relationship of the Mie scattering relative flux and the measurement ratios of DRX and PDR to TEOM ($R^2$=0.95 for DRX and 0.90 for PDR) shown in Fig. 2, verifies the linear relationship between RF and the ratio of optical instruments to reference instruments for this study, and provides an explanation of the performance of DRX and PDR for the variation in particle composition and size. The particle with a higher refractive index (such as $(NH4)_2SO_4$ and sucrose) or larger size distribution, would produce the larger

relative flux for optical instruments, resulting in higher instrument response and subsequently a higher measurement ratio of optical sensor readings to reference values.

**3.3 Detection efficiency of OPC-N2**

To focus on the counting ability of OPC-N2 in more detail, the detection efficiency in the first size range channel (380-540 nm) of the OPC-N2 was analyzed. The limited size range was determined using the overlap

of SMPS size range (14.9-673.2 nm) with OPC (detection limit of 380 nm). Using the measurements from the OPC and SMPS size distributions, the particle number counts of these two instruments were calculated in the overlapping size range, in units of number per $cm^3$. The comparison of OPC-N2 and SMPS measurement values is shown in Fig. 3. In the analysis, $(NH_4)_2SO_4$ readings showed saturation when OPC counts were higher than 300 # $cm^{-3}$. Excluding the saturated data, all four compounds ($(NH_4)_2SO_4$, $NaNO_3$, sucrose, adipic

acid) displayed strong linear relationships between SMPS and OPC-N2 counts, with all $R^2$>=0.99.

As with the DRX and PDR, the detection efficiency of the OPC-N2 showed substantial differences with aerosol composition. For $NaNO_3$, the OPC-N2 detection efficiency was as high as 103% when compared to the SMPS. However, it was only 42% for $(NH_4)_2SO_4$, 55% for sucrose, and 16% for adipic acid. The low detection efficiency of the OPC-N2 for many of these compounds helps explain the low ratio of the OPC to



SMPS in Table 3. These limited results also indicate one of the complexities in assessing the performance of the OPC-N2.

**3.4 Ambient measurement**

Ambient measurements were recorded from 12/22/2016 to 01/07/2017, and from 09/27/2017 to 10/01/2017 to

study the performance of PDR and DRX under ambient aerosol conditions. The time series of 1-hour aerosol concentration readings of the DRX, PDR, AMS, SMPS are shown in Fig. 4, and the aerosol chemical composition mass concentration with aerosol mass fraction are shown in Fig. S5.

Figure 4 shows very similar behavior (that is, high and low excursions) for the DRX, PDR, SMPS, and AMS measurements. AMS measurements showed high correlation ($R^2$=0.94) with SMPS mass concentration, with

slope = 0.85 as shown in Fig. S6a. SMPS values ware about 15% lower than AMS, which may be the result of the bias of the estimated density or the CE factor used by AMS. The high correlation verifies the performance of AMS for these periods and its reliability as a reference instrument. The even higher coefficient of determination ($R^2$=0.96) between PDR and DRX (Fig. S6b) shows the similarity of these two optical instruments, and the slope of 1.81 indicates the different factory calibration factor for these two optical

instruments. Of all the DRX vs PDR data in Fig. S6b, there was one set of data points clearly deviating from the main cluster, and these data points came from 9/28/2017 02:00 to the end of this study. At that date and time, a cold front passed through Albany, causing aerosol mass concentration to drop quickly, from about 11 ug m$^{-3}$ to 2 ug m$^{-3}$, and aerosol mass median diameter (measured by SMPS) dropped from 280nm to 200nm while the organic compound fraction increased to above 80%, as shown in Fig. S5. The difference between

the slopes of these deviated data points (shown in red, slope=2.45) and the slope of main cluster to the left (shown in black, slope=1.89) shows the influence of aerosol characteristics (mainly aerosol size here) on these two optical instruments. The comparison between the optical instruments with AMS is shown in Fig. 5. PDR data shows more scatter than DRX ($R^2$ 0.86 vs. 0.91), and the slope of PDR to AMS was 1.03 (near to 1), while the slope of DRX to AMS was 1.96, indicating a calibration factor of 0.52 would be necessary for this

study, which is higher than the calibration factor recommended by the manufacturer for ambient aerosol (0.38,





TSI Inc., 2017). This difference in recommended calibration factor can plausibly be caused by different kinds of aerosol during the different studies performed.

To investigate the dependence of the optical sensors on aerosol size, four periods are separated from the whole time series, based on aerosol mass median diameter (Fig. S5). These are Period_1: 12/22/2016 00:00 to

01/05/2017 23:59 with mean median diameter of 249 nm; Period_2: 01/06/2017 00:00 to 01/08/2017 10:00 with mean median diameter of 219 nm; Period_3: 09/27/2017 09:30 to 09/28/2017 02:00 with mean median diameter of 258 nm; and Period_4: 09/28/2017 02:00 to 10/01/2017 14:00 with mean median diameter of 192 nm). For each optical instrument, the highest slope (Fig. S7) occurs during period_3 (slope=1.70 for PDR, and 2.9 for DRX), and was 70% (PDR) to 48% (DRX) higher than the overall average (1.03 for PDR, and 1.96 for

DRX) shown in Fig. 5, while the lowest slopes happened during period_2 and period_4. Combining the averaged median diameters of four periods with the normalized volume distribution (Fig. S8), we can see once again that particles with larger size would result in a higher optical instrument response.

The combination of aerosol median diameter, aerosol compound mass fractions, and the ratio of optical instrument values to AMS mass concentrations can be used to verify the above assumption that the aerosol

size is the most important variable in this study. Plots combining these parameters are shown in Fig. 6 (PDR) and Fig. S9 (DRX). Figure 6 shows the correlation scatterplot of aerosol median diameter with the PDR/AMS ratio. All points are color-coded by organic mass fraction, and sized by AMS mass concentration. Moderate coefficients of determination ($R^2$=0.64) with positive slopes indicate the innate relationship between mass median diameter and PDR/AMS ratio. The random distribution of organic-rich (red) points and inorganic-rich

(blue) points in Fig. 6 and Fig. S9 suggests aerosol composition has a smaller effect on the response of these optical instruments. For example, the organic mass fraction corresponding to ratio = 1 ranged from 0.3 to 0.8 in Fig. S9. Figure 6 still shows clustered organic-rich particles in the smaller size range (lower left points), as well as inorganic-rich particles in the large size range (upper right points). A likely reason for this is that the small size range particles (such as ones in Period_4) were newly-emitted fresh aerosol that was characterized

by very high organic fraction, small-size organic distribution and externally- mixed properties (note that organics had a small second peak at about 150nm, Fig. S10a, Sun et al., 2009), while the large particles (such



as ones in Period_3) were related to long-range transported aged aerosol, characterized by higher $SO_4$ mass fraction, large-size organic/$SO_4$ mass distribution as well as internally-mixed properties (both peak about 400 nm, Fig. 10b, Sun et al., 2009).

The dependence of the optical sensors on aerosol size highlights an important consideration for the use of optical scattering sensors in critical applications. It is clear that different correction factors should be used in different measurement conditions instead of a single constant value (McNamara et al., 2011). For example, the PDR, when used in a rural forest environment with high concentrations of fresh organic-rich small size aerosol, the correction factor may be as high as 2.50 based on the results of this study, while for an area which would be strongly impacted by aerosol transported long distances, such as the northern U.S. regions affected by long-range transported wood-fire produced aerosol from western Canada (Le et al., 2014) or biomass-burning aerosol from of the central U.S. (such as the Mississippi Valley, Zhang et al., 2008), the correction factor could be as low as 0.60, such as in period_3, when Albany likely was affected by biomass-burning aerosol from southern Mississippi Valley as shown in Fig. S11.

### 3.5 Ambient aerosol refractive index estimation

Assuming the relationship between RF and the ratio of optical instruments to reference instruments is constant for lab tests of pure composition aerosols (Fig.2) and ambient aerosol, the average ambient aerosol refractive index real part can be derived. The averaged RF of optical instruments would be calculated based on that linear relationship and the ratio of optical instruments to AMS. Furthermore, a reference table of RF for different refractive index can be built based on the normalized volume size distribution and the assumed differences in refractive index (from 1.2 to 1.8 with step of 0.01), as shown in Table S3. After comparing the calculated RF with the RF in the reference table, the refractive index may be derived. Figure S12 shows the estimated time series of 1-h refractive index real part using the above method based on PDR and DRX data. Generally, the relative difference between these two estimations was below 10%, with the largest discrepancy during period_4. A likely explanation of this larger difference is the smaller particles in period 4 biasing the optical instruments relative response. The averaged value determined for the refractive index was 1.54 for PDR and 1.55 for DRX, which was very near to the estimated value 1.56 (Hand et al., 2002), and within the





estimated range of 1.54 to 1.72 (Ebert et al., 2004). The correlation scatter plot of aerosol refractive index and

the PDR/AMS ratio (Fig. S13a) verifies the smaller effect of refractive index on PDR/AMS ratio compared to

aerosol size, with a similar result for the DRX (Fig. S13b). One possible reason for this is that the range of

variation of refractive index of this study was relatively small (88% points in 1.48-1.58), which was not

enough to cause significant variation in the optical instrument response.

**4 Conclusion**

In this study, the performance of three optical sensors (DRX, PDR, and OPC-N2) was evaluated using 1)

poly- and mono-disperse aerosol in the lab, and 2) ambient aerosol (PDR and DRX only). The aim of this

evaluation was to study the applicability and limitations of each optical sensor. A Mie scattering calculation

was used to describe the results of these measurements. During laboratory tests, good linear relationships

(generally $R^2>0.90$) were shown between the optical measurements and the traditional mass measurements,

while the slope depended on aerosol size and aerosol composition. The response of these optical instruments

can be well explained by the Mie scattering calculations. During the mono-disperse particle tests, the DRX

was more sensitive to smaller particles than PDR, which is consistent with its shorter wavelength light source.

During the poly-disperse particle experiments, all three sensors showed higher responses for sucrose and

$(NH_4)_2SO_4$, and lower responses for $NaNO_3$ and adipic acid, which illustrates the important effect of

refractive index (or particle chemical composition) on instrument performance. The aerosol with higher

refractive indices or larger size produced more scattering flux, and therefore a higher instrument response.

During ambient aerosol experiments, the DRX and PDR were directly correlated to the reference instruments

(SMPS and AMS). By exploring the aerosol mass median diameter measured by SMPS and combining the

mass fraction loading of aerosol compounds measured by AMS, we found aerosol size (represented by

aerosol mass median diameter) has the greatest impact on the data in this study when compared to the

chemical composition of the aerosol compounds and the aerosol refractive index. The aerosol refractive index

was estimated based on the relationship of RF with the ratio of optical instruments to reference instruments,

the normalized volume size distribution, and a reference table.



The dependence of the optical sensors on aerosol size highlights an important consideration about aerosol size distribution in the use of optical scattering sensors. For field ambient aerosol measurements, the characteristics of aerosol sources, such as traffic emissions or forest-based new particle formation may effect on the quality of the sensor data. The general size distribution of aerosol from near constant sources from

former studies would help to determine more accurate calibration factor for optical instruments. However, due to the limit of SMPS measurements (upper size limit <700 nm) and AMS measurements (upper size limit <1000 nm, and only detects non-refractory species), short time period measurements of this study, and a lack of diversity of aerosol sources, more ambient measurements will be necessary to better ascertain the application of optical instruments. Despite the complexity of determining calibration factor, as well as

instrument limitations, these compact optical instruments will hopefully provide increasingly reliable data covering a greater spatial extent. Additional studies and measurements will help better characterize the aerosol, and it is hoped they will provide further accurate information that will help inform and design plans to improve ambient air quality.

Acknowledgements: This work has been supported by the New York State Energy Research and Development Authority (NYSERDA) contract number 48971. Special thanks go to research scientist Dr. Patricia Fritz (New York State Department of Health) for granting use of the PDR, Xiuli Wei and Hui Shi for helping in the lab work, and Brian Crandall for technical writing of this paper.

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





**Table 1.** The scattering angles and light source wavelengths used in the DRX, PDR, and the OPC-N2.

| Sensor | DRX | PDR | OPC-N2 |
|---|---|---|---|
| Wavelength (nm) | 655 | 880 | 658 |
| Scattering angle (°) | 90 | 80-150 | 30 |

**Table 2.** Slopes of the regression lines obtained when plotting optically reported PM values (for DRX, PDR, and OPC-N2); and calculated Mie scatter flux (for DRX_RF and PDR_RF) for different PSL sizes versus the PM mass concentration measured by the TEOM.

| Vs TEOM | 90 nm | 173 nm | 304 nm | 490 nm | 1030 nm |
|---|---|---|---|---|---|
| DRX | 0.86 | 0.90 | 3.73 | 2.56 | 0.93 |
| PDR | 0.32 | 0.28 | 1.34 | 3.14 | 1.30 |
| OPC-N2 | N/A | N/A | N/A | 0.68 | 0.48 |
| RF results | | | | | |
| DRX_RF | 0.53 | 3.73 | 11.07 | 8.90 | 4.83 |
| PDR_RF | 0.09 | 0.63 | 2.92 | 5.30 | 3.70 |

**Table 3.** Ratios of mass concentration measured by optical instruments to reference instruments for the four compounds for the groups in Table S1.

| Ratio (vs TEOM or SMPS) | | $(NH_4)_2SO_4$ | $NaNO_3$ | sucrose | adipic acid* |
|---|---|---|---|---|---|
| Group 1 | DRX | 0.85 | 0.58 | 0.92 | 0.61 |
| | PDR-1500 | 0.32 | 0.21 | 0.32 | 0.24 |
| | OPC-N2 | 0.026 | 0.015 | 0.018 | 0.01 |




| | | | | | |
|---|---|---|---|---|---|
| Group 2 | DRX | 1.34 | 0.96 | 1.65 | 0.91 |
| | PDR-1500 | 0.66 | 0.40 | 0.61 | 0.39 |
| | OPC-N2 | 0.09 | 0.07 | 0.08 | 0.03 |
| Group 3 | DRX | 1.57 | 1.06 | 1.90 | 1.23 |
| | PDR-1500 | 0.88 | 0.62 | 1.08 | 0.62 |
| | OPC-N2 | 0.2 | 0.14 | 0.17 | 0.09 |

*SMPS was used as the reference measurement for adipic acid (see text).

**Table 4.** Refractive indices of each aerosol used in the Mie scattering calculations

| | $(NH_4)_2SO_4$ | $NaNO_3$ | Sucrose | adipic acid |
|---|---|---|---|---|
| n | 1.521/1.523/1.533 | 1.587/1.336 | 1.54/1.567/1.572 | 1.439 |



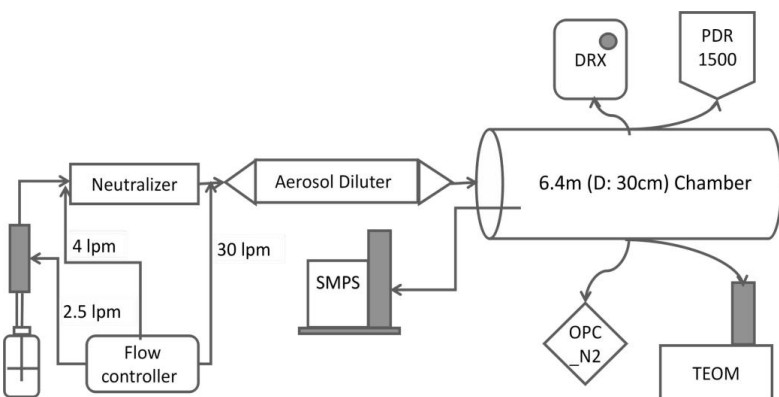

**Figure 1: Laboratory setup used for the evaluation and calibration of optical instruments using poly- and mono-disperse test aerosol**.

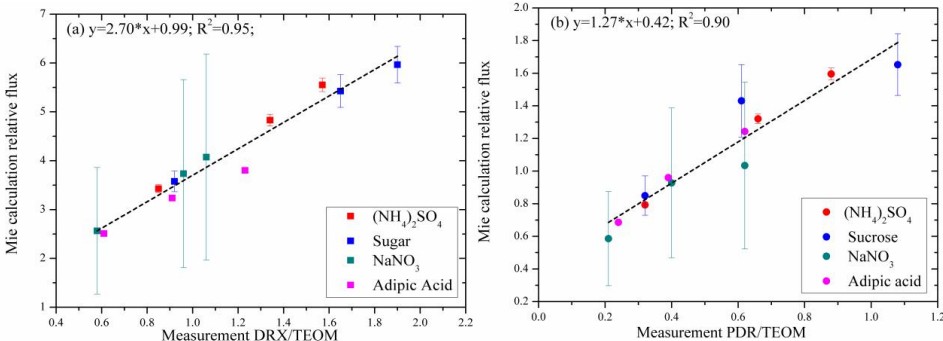

**Figure 2: The relationship of the Mie scattering relative flux and the measurement ratio of optical sensors to TEOM, (a) DRX; (b) PDR.**



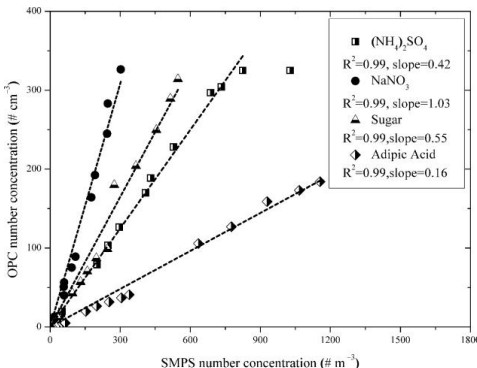

**Figure 3: The comparison of OPC and SMPS counts for four kinds of aerosol in the 380-540nm size range.**

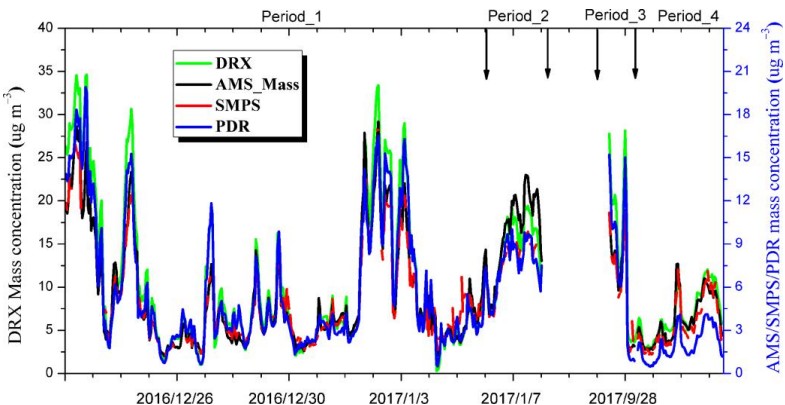

**Figure 4: The time series of 1-hour aerosol concentration of DRX, PDR, AMS, SMPS from 12/22/2016 to 01/07/2017 and 09/27/2017 to 10/01/2017.**



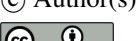

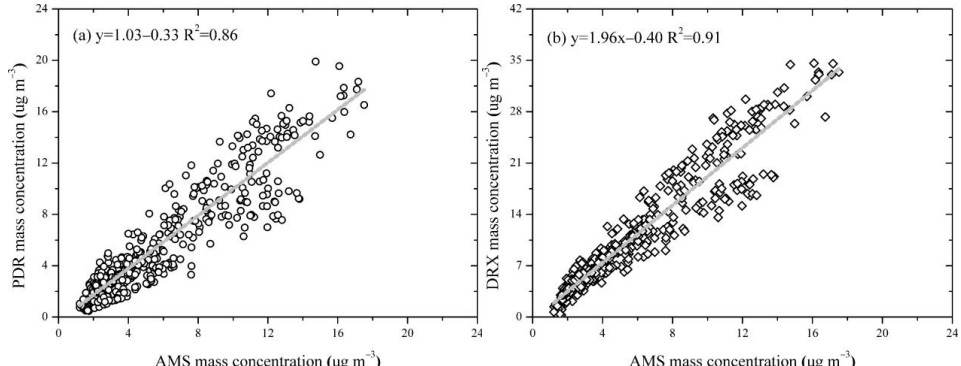

**Figure 5: The comparison of the 1-hr PM₂.₅ average concentration between PDR (a) and DRX (b) observations with AMS measurements.**

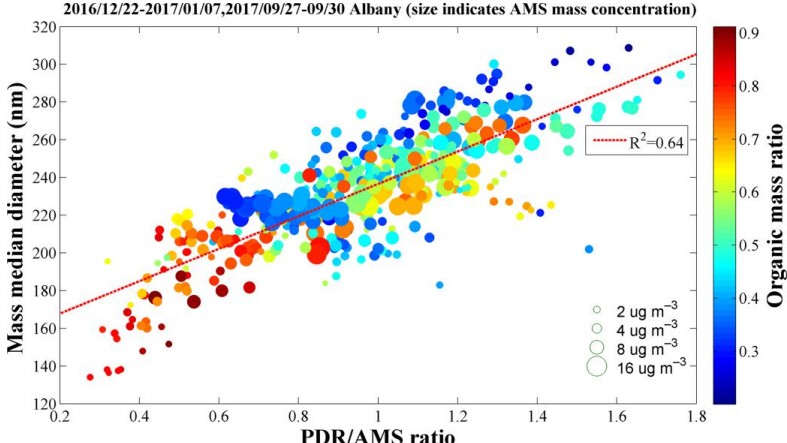

**Figure 6: The correlation scatterplot of aerosol median diameter and the PDR/AMS ratio, and all points are colored by organic mass fraction, and sized by AMS mass concentration.**