# Peer review of "Exploring the Applicability and Limitations of Selected Optical Scattering Instruments for PM Mass Measurement"

_Atmospheric Measurement Techniques, 2017_

## Referee Comment (RC1) · Anonymous Referee #1 · 22 Feb 2018

The manuscript by Zhang et al. evaluated three optical sensors through both laboratory and field measurements. The results show a large variability in response for these optical instruments depending on aerosol size and compositions. In recent years, the portable optical instruments are becoming more and more important in aerosol particle measurements, yet the performance is not fully evaluated yet. This study is critical for future users by providing important information on measurement uncertainties. This manuscript is overall well written, and I recommend it for publication after minor revisions.

Comments:

[Figure]

1. Suggest describing briefly the major conclusions how aerosol size, composition and refractive index influence optical response in the abstract.

2. Abstract, line 13 – 15, rephrase. It is difficult to read.

3. Page 2, second paragraph, suggest adding some discussions on vertical measurement to highlight the advantage of portable scattering instruments.

4. Page 4, line 2, make sure the density of PSL. It is 1.05 g cm-3.

5. Page 5, line 7, any reasons for choosing these four different compounds?

6. Use log scale for x-axis in Figure S3.

---

## Referee Comment (RC2) · Anonymous Referee #2 · 26 Feb 2018

The manuscript presents an evaluation of three low-cost, optical-based aerosol sensors by comparing their response to that of several different instruments to laboratory-generated and ambient aerosol. The sensors generally showed good correlations with different reference instruments, but different relationships (slopes), due to differences in sensor properties, calibration, and sampled aerosol properties. The topic is important given the recent interest in using these types of sensors for air quality monitoring, and is appropriate for AMT. In current form, however, the manuscript lacks in a few areas, as described below, and requires significant revision before I can recommend its publication.

General comments:

Considering the paper focuses on an evaluation of several sensors I think much more detail regarding the sensors should be provided. For example, sample flow rates, whether or not a sheath flow is used, lower and upper concentration limits, etc. Additional information on how exactly are calibrations performed by manufacturer should be included as well.

PSL is one of the primary test particle types, but in my experience it can be very difficult generating a pure monodisperse size distribution from PSL without first passing it through a DMA. When PSL is atomized/nebulized it can produce a large mode of fine particles with sizes < 200 nm, which arises from surfactant and other contaminants in the PSL solution. These smaller particles can affect mass measurements depending on the lower detection limit of the sensor. Do the authors have any SMPS measurements showing the quality of the PSL output and confirming that it was limited to just the PSL at the expected diameters?

Tests were only performed on dry aerosol, which is probably necessary to eliminate the complications introduced by water uptake on particles. The manuscript would benefit from some discussion of this however, in that low-cost sensors will need to either account for RH impacts or be actively dried when sampling.

Finally, I think the manuscript is significantly weakened by the lack of a true reference measurement for the ambient measurements, having to rely instead on AMS and SMPS measurements, neither of which is a true mass measurement. The authors' give a reason for this, however I find it somewhat puzzling that the TEOM was unable to provide quality measurements of ambient PM mass.

Finally, there are few places where the wording / grammar is awkward and would benefit from copy editing.

Specific comments:

2, 18: recommend including a statement that the PAS is also an optical particle counter and not a direct mass measurement

2, 20: there are several additional, but quite new references that could be included and compared to here, such as Crilley et al. (2018) and a paper currently in discussion (Feinbeg et al., 2018).

3, 4-5: change to "aerosol mobility diameter" and to "non-refractory chemical composition"

3, 14 & 20: Please provide the scattering angles measured for the DRX and PDR

3, 25: are the specifications more specific? A D50 would be helpful if provided.

4, 2: more appropriate to state that the conversion is done on volume concentrations, not number concentrations

5, 1: please provide typical RH in the chamber for experiments

5, 17: While it may be true that the OPC-N2 does not measure below 380 nm, it would have still been useful to see the level of disagreement. I'm also a little surprised the TEOM could not be used due to the high-frequency noise, since this device is designed for this type of monitoring application. Could a longer averaging period have been used?

7, 21: Clarify that the 0.45 g/L refer to the nebulizer solution concentration. It would be good to state the density of sucrose somewhere (1.59 g / cm3).

7, 20-22: It is strange that the OPC-N2 shows such good agreement for the sucrose in this example. It's stated elsewhere it has a poor detection efficiency, especially below its lower limit (380 nm). Figure S3a shows aerosol volume distributions, which show at least half of the distribution below 400 nm. This would suggest the OPC-N2 misses at least half of the aerosol volume (and therefore mass), yet it looks very similar to the other two sensors, and shows good agreement with the TEOM. How is this possible?

The figure is also not consistent with Table 3, which lists very low ratios, as expected.

Also, Figure S4's legend shows PDR * 3. Does this mean the data for that sensor are multiplied by 3? I don't think this is discussed in the text. More details should be provided justifying this if so.

7, 23: The results suggest good linearity, but I'm not convinced on stability. Were tests repeated after some period of time, and simulated usage/exposure to ambient air? That type of test would be needed to verify stability of the response. If the author's mean stability since original calibration more details on the calibration should be given, including dates, and what was done with sensors between the time of calibration and the testing.

8, 2: Is the adipic acid result normalized against the TEOM or AMS? Not clear here, but earlier it's stated the compound is too volatile to use the TEOM.

9, 3: The OPC-N2 provides binned results though, correct? These would have some dependence on the pulse intensity, which is related to refractive index as well as size, shape.

9, 17: Stating this overlap range explicitly here would be helpful.

9, 22: Is this section comparing number detection efficiency or mass efficiency? Please be clear.

10, 1-2: What is meant by this statement? The sensor response to these compounds due to the limited detection range of the OPC-N2 is poor, but that does not mean the evaluating its performance is complex.

10, 10: Another factor not mentioned is mass from species not measured by AMS, including BC, dust, etc.

10, 11-12: While the SMPS and AMS are consistent, I think the argument that this means the AMS is a good reference instrument for mass is somewhat weak. This is

where the analysis suffers from not including the TEOM, a true PM mass reference measurement method.

10, 14: This seems like a large difference between two instruments both calibrated against the same material (Arizona test dust)

11, 1-2: This statement needs to be supported by more evidence. What constitutes "ambient aerosol" according to TSI? Why is the difference plausible? It seems like the aerosol composition at the test site is not radically different from what would be considered a regional, continental aerosol (as opposed to say, a heavily marine influenced location).

11, 4-12: It seems like the more straight-forward way to evaluate this relationship would be to plot ratios of each sensor to the reference as a function of mass mean diameter measured by the SMPS, as shown in Figure 6. Suggest removing.

11, 16: I suggest swapping the axes to show mean diameter on the x-axis.

12, 26: where do these estimated RI come from? Are they based on the aerosol composition?

Table 1: Suggest adding rough cost of each instrument as I think it is relevant in the context of the manuscript.

Table 3: It would help reader interpret the different groups if you listed the median diameter or similar parameter under the group name. Information in Table 4 could combined here and eliminate a table.

References

Crilley, L. R., et al., Evaluation of a low-cost optical particle counter (Alphasense OPC-N2) for ambient air monitoring, AMT-11-709-2018.

Feinberg, S., et al., Long-term evaluation of air sensor technology under ambient conditions in Denver, Colorado, ATMD-2018-12

---

## Author Response (AR1)

**Response to Reviewer 1 comments**

We thank the reviewer for her/his comments, and have carefully addressed each comment and improved the paper. Below find our point-by-point responses to reviewer 1 comments, where first the reviewer comment is repeated in italics, followed by our response in bold.

5    Comments to the Author:

*1. Suggest describing briefly the major conclusions how aerosol size, composition and refractive index influence optical response in the abstract.*

**We have substantially rewritten the abstract to address this suggestion.**

*2. Abstract, line 13 – 15, rephrase. It is difficult to read.*

10   **This sentence was completely rewritten as described above.**

*3. Page 2, second paragraph, suggest adding some discussions on vertical measurement to highlight the advantage of portable scattering instruments.*

**Thanks for this comment. We added one sentence to describe advantage of portable scattering instruments for vertical profile measurements.**

15   *4. Page 4, line 2, make sure the density of PSL. It is 1.05 g cm-3.*

**The 1.65 g cm$^{-3}$ is the value used in the OPC-N2 software for transferring the volume concentration of ambient aerosol to mass concentration, and it is not the density of PSL.**

*5. Page 5, line 7, any reasons for choosing these four different compounds?*

**Compounds were chosen that first of all were available to us in our laboratory. More importantly,**
20   **these compounds represented some of the major chemical species types (sulfate, nitrate, ammonium, organics) that we observed in the northeast US. Third, the compounds were water soluble and relatively stable, so that we could generate well characterized pure synthetic aerosols.**

*6. Use log scale for x-axis in Figure S3.*

**Thanks for this comment. As you suggestion, we modified the figure S3.**

25   Again thank you for all your valuable comments, which have helped to improve the paper.

Jie Zhang, Joseph P. Marto, James J. Schwab

**Response to Reviewer 2 comments**

We thank the reviewer for her/his comments, and have carefully addressed each comment and improved the paper. Below find our point by point responses, where first the reviewer 2 comment is repeated in italics, and is followed by our response in bold.

Comments to the Author:

**General comments:**

*Considering the paper focuses on an evaluation of several sensors I think much more detail regarding the sensors should be provided. For example, sample flow rates, whether or not a sheath flow is used, lower and upper concentration limits, etc. Additional information on how exactly are calibrations performed by manufacturer should be included as well.*

**Thanks for this comment. We have added the more detailed description of the sensors into the text in section 2.1, as well as updated the information of the calibration aerosol for these instruments.**

*PSL is one of the primary test particle types, but in my experience it can be very difficult generating a pure monodisperse size distribution from PSL without first passing it through a DMA. When PSL is atomized/nebulized it can produce a large mode of fine particles with sizes < 200 nm, which arises from surfactant and other contaminants in the PSL solution. These smaller particles can affect mass measurements depending on the lower detection limit of the sensor. Do the authors have any SMPS measurements showing the quality of the PSL output and confirming that it was limited to just the PSL at the expected diameters?*

**You are right and the issue is known to the authors. During our test, there were a large mode of fine particles with number size < 100nm, as shown in Figure 1a (304nm PSL for example). Fortunately the volume concentration distribution mainly concentrated around 304nm (Fig 1b). For the DRX (with the shorter wavelength light source) , the Mie calculation result of the full volume distribution is only about 6% different than the calculation using only the 304 nm PSL. Based on this calculation, we used the simplification of only considering the factory size value of PSL for our Mie scattering calculation.**

[Figure]

**Figure 1. Left: One sample of the lognormal number concentration distribution of 304nm PSL; Right: the volume concentration distribution of the 304nm PSL.**

*Tests were only performed on dry aerosol, which is probably necessary to eliminate the complications introduced by water uptake on particles. The manuscript would benefit from some discussion of this however, in that low-cost sensors will need to either account for RH impacts or be actively dried when sampling.*

**Thank you for this good suggestion. Actually, during lab measurements, the RH detected by PDR is below than 10%, and during ambient measurements, a silica dryer was used for keeping the RH below 40%, which is required by AMS. Based on the fact that our tests only focused on dry aerosol, we add the shortcoming of the result in the final paragraph of the conclusion.**

*Finally, I think the manuscript is significantly weakened by the lack of a true reference measurement for the ambient measurements, having to rely instead on AMS and SMPS measurements, neither of which is a true mass measurement. The authors's give a reason for this, however I find it somewhat puzzling that the TEOM was unable to provide quality measurements of ambient PM mass.*

**Actually, we had the TEOM ambient aerosol mass concentration from 12/22/2016 to 01/07/2017. Its relationship with AMS 1-hour data is shown in below Figure 2 (a) and (b). Figure 2(a) shows similar trends for TEOM and AMS, and at the same time, there is only fair ($R^2$ of 0.41 and a slope of 0.88). From figure (a), our TEOM showed higher noise than the AMS (and the other instruments), and the possible reasons maybe the flow tube which connect the TEOM to the aerosol inlet is too long. At the same time, AMS and SMPS were self-verified through estimating the aerosol density (which is needed by the SMPS) through AMS compounds measurements (Qi et al., 2005). Also our TEOM did not work from 09/27/2017 to 10/01/2017. More importantly is the very good association of the DRX, PDR, SMPS, and AMS shown in Figure 4. Clearly the TEOM is the "odd man out" in this array of measurements. Had the TEOM measurements been better quality, we would**

**have happily used them as our reference for the ambient data. Based on the above consideration, we used AMS and SMPS as reference instruments.**

[Figure]

(a)              (b)

**Figure 2. (a): The time series of 1-hour aerosol concentration of TEOM and AMS from 12/22/2016 to 01/07/2017 and 09/27/2017; (b) the comparison of the 1-hr PM2.5 average concentration between TEOM and AMS**

**Specific comments:**

*2, 18: recommend including a statement that the PAS is also an optical particle counter and not a direct mass measurement*

**Thank you for this comment. We had added this information into the text.**

*2, 20: there are several additional, but quite new references that could be included and compared to here, such as Crilley et al. (2018) and a paper currently in discussion (Feinbeg et al., 2018).*

**Thank you for this comment. We had added these references into the text.**

*3, 4-5: change to "aerosol mobility diameter" and to "non-refractory chemical composition"*

**Thank you for this comment. We had corrected the expression in the text.**

*3, 14 & 20: Please provide the scattering angles measured for the DRX and PDR*

**Thank you for this comment. We had added the information of the scattering angles into the text.**

*3, 25: are the specifications more specific? A D50 would be helpful if provided.*

**Thank you for this comment. We corrected the word "specifications" to "manual". D50 data is not provided for the PDR nor the DRX. Both instruments list a "particle size range" between 0.1 and 10 microns, and this information has been added to the text.**

*4, 2: more appropriate to state that the conversion is done on volume concentrations, not number concentrations*

**Thank you for this comment. We corrected the expression in the text.**

*5, 1: please provide typical RH in the chamber for experiments*

**Thank you for this comment. We had added the information of the RH into the text (page 5, 20)**

*5, 17: While it may be true that the OPC-N2 does not measure below 380 nm, it would have still been useful to see the level of disagreement. I'm also a little surprised the TEOM could not be used due to the high-frequency noise, since this device is designed for this type of monitoring application. Could a longer averaging period have been used?*

**During the ambient measurement periods, the value of OPC-N2 was generally near to 0, so there was little point in reporting it. The problem with the TEOM is discussed above. For our TEOM, as described in the general comments section, high-frequency noise was shown, and maybe due to the flow tube which connect the TEOM and aerosol inlet was too long. Longer period would work, while short averaging period data would help us to better study the performance of DRX and PDR, so in the text, we used AMS and SMPS as reference instruments**

*7, 21: Clarify that the 0.45 g/L refer to the nebulizer solution concentration. It would be good to state the density of sucrose somewhere (1.59 g / cm³).*

**Thank you for this comment. We had added the clarification into the text .**

*7, 20-22: It is strange that the OPC-N2 shows such good agreement for the sucrose in this example. It's stated elsewhere it has a poor detection efficiency, especially below its lower limit (380 nm). Figure S3a shows aerosol volume distributions, which show at least half of the distribution below 400 nm. This would suggest the OPC-N2 misses at least half of the aerosol volume (and therefore mass), yet it looks very similar to the other two sensors, and shows good agreement with the TEOM. How is this possible?*

**Pretty much every optical sensor (and even more so for the inexpensive ones used in very low cost applications) only respond to the tail of the volume size distribution. If the shape of the size distribution is "well behaved" (or well characterized) the response can be quite linear with**

**increasing concentration. That quite well explains the OPC results. The paper takes pains to point out that when the size distribution or chemical composition is changing, one needs to take care with optical measurements.**

*The figure is also not consistent with Table 3, which lists very low ratios, as expected. Also, Figure S4's legend shows PDR * 3. Does this mean the data for that sensor are multiplied by 3? I don't think this is discussed in the text. More details should be provided justifying this if so.*

**Thank you for this comment. Table 3 illustrates that the DRX response per unit mass for the particle groups tested was consistently highest, followed by the PDR, and both are much higher than the OPC. The multiplicative factors of 3 for the PDR and 50 for the DRX reflect these relative responses, and the caption of Figure S4 is much more explicit about these factors. The main point of Figure S4 is the linearity of the responses – the measured values are in Table 3.**

*7, 23: The results suggest good linearity, but I'm not convinced on stability. Were tests repeated after some period of time, and simulated usage/exposure to ambient air? That type of test would be needed to verify stability of the response. If the author's mean stability since original calibration more details on the calibration should be given, including dates, and what was done with sensors between the time of calibration and the testing.*

**Thank you for this comment. Lab tests were repeated numerous times, always with nearly identical results. The ambient tests were performed 9 months apart, again with similar results. The optical sensors are quite stable as long as the sampled aerosol itself is stable.**

*8, 2: Is the adipic acid result normalized against the TEOM or AMS? Not clear here, but earlier it's stated the compound is too volatile to use the TEOM.*

**Thank you for this comment. The adipic acid result was normalized against the SMPS, and the statement of adipic acid calculation was stated in Page 6, 5, as well as in Table 3.**

*9, 3: The OPC-N2 provides binned results though, correct? These would have some dependence on the pulse intensity, which is related to refractive index as well as size, shape.*

**Thank you for this comment. The binned results of OPC-N2 are reported as pulse intensity, and the relationship between the calculated Mie scattering relative flux and the measurement ratio of OPC-N2 to SMPS in OPC-N2 bin range 380-540nm is shown in Figure 3. There is a linear relationship between the calculated Mie relative flux and the ratio of OPC-N2 to reference instrument (SMPS), which indicates that the OPC-N2 binned pulse intensity is related to refractive index as well as size. Based on the consideration that OPC-N2 first transfers the pulse intensity to number, then transfers**

**to mass concentration, which means that the scattering flux is not directly converted to reported the mass concentration, we did not including this result into the Mie calculation section. At the same time, we corrected our former expression in the text to ensure a more precise statement to this effect.**

[Figure]

**Figure 3 The relationship of the Mie scattering relative flux and the measurement ratio of OPC-N2 to SMPS**

*9, 17: Stating this overlap range explicitly here would be helpful.*

**Thank you for this comment. We added the size range into the text.**

*9, 22: Is this section comparing number detection efficiency or mass efficiency? Please be clear.*

**Thank you for this comment. We corrected the confusing expression, and add "number" into the text.**

*10, 1-2: What is meant by this statement? The sensor response to these compounds due to the limited detection range of the OPC-N2 is poor, but that does not mean the evaluating its performance is complex.*

**Thank you for this comment. We deleted that confusing expression.**

*10, 10: Another factor not mentioned is mass from species not measured by AMS, including BC, dust, etc.*

**Thank you for this comment. During the analyzing, SMPS was lower than AMS value. Though AMS did not measure BC and dust, adding this statement would larger the difference between SMPS and AMS. So we keep the origin statement that " SMPS values ware about 15% lower than AMS,**

**which may be the result of the bias of the estimated density or the CE factor used by AMS".**

*10, 11-12: While the SMPS and AMS are consistent, I think the argument that this means the AMS is a good reference instrument for mass is somewhat weak. This is where the analysis suffers from not including the TEOM, a true PM mass reference measurement method.*

**Thank you for this comment. As noted above, a well operating TEOM with low noise would have been useful to the study. However, in addition to the reasoning above, the TEOM versus AMS measurements from numerous field and lab studies (including the lab studies presented in this paper) give us high confidence in the AMS as a good mass measurement for non-refractory submicron particles. Since this type of particle encompasses the vast majority of what we measured, we are confident in using the AMS for this purpose.**

*10, 14: This seems like a large difference between two instruments both calibrated against the same material (Arizona test dust)*

**The difference from the difference of wavelength used by these two instruments. As shown in lab tests, these two instruments show a big difference for same aerosol distribution. So though they are calibrated by same material, they can respond differently for ambient aerosols in situations with quite different size distributions. A second difference is the calibration factors used. The two instruments recommend different calibration factors for "ambient" measurements.**

*11, 1-2: This statement needs to be supported by more evidence. What constitutes "ambient aerosol" according to TSI? Why is the difference plausible? It seems like the aerosol composition at the test site is not radically different from what would be considered a regional, continental aerosol (as opposed to say, a heavily marine influenced location).*

**The calibration factor 0.38 came from Wallace's result (Wallace et. al., 2011), which tested the DRX for ambient air in one location. A complete characterization requires data for many different kinds of aerosol, including aerosol composition as well as size distribution for Wallace's study and our study. From the TSI "APPLICATION NOTE EXPMN-007", there were still different calibration factors for ambient air, and TSI mainly suggests the one from Wallace's result. For the aerosol composition from regional, continental vs. Marine location, they would be different, and new calibration factors are still needed.**

*11, 4-12: It seems like the more straight-forward way to evaluate this relationship would be to plot ratios of each sensor to the reference as a function of mass mean diameter measured by the SMPS, as shown in Figure 6. Suggest removing.*

**Thank you for this comment. After considering carefully, we think it is truly not necessary. So we deleted the whole paragraph.**

*11, 16: I suggest swapping the axes to show mean diameter on the x-axis.*

**Thank you for this comment. We use "DRX(PDR)/AMS ratio" as "x axes" to emphasize the performance of these sensors, and give readers more directly impression. We would keep the original use.**

*12, 26: where do these estimated RI come from? Are they based on the aerosol composition?*

**The refractive index is estimated based on the above method mentioned in section 3.5 using the reference table.**

*Table 1: Suggest adding rough cost of each instrument as I think it is relevant in the context of the manuscript.*

**Thank you for this comment. We had added the price information into text (Table 1).**

*Table 3: It would help reader interpret the different groups if you listed the median diameter or similar parameter under the group name. Information in Table 4 could combined here and eliminate a table.*

**Thank you for this comment. We added the median diameter below different groups, and combining Table 3 and Table 4 into one. Thank you so much!**

Again thank you for all your valuable comments, which have helped to improve the paper.

[revised manuscript text omitted]
 (For convenient comparison, PDR data is multiplied by 3, while OPC-N2 data s multiplied by 50).**

[Figure]

[Figure]

**Figure S5. (a) A time series of 1-hour aerosol composition concentration and SMPS mass median diameter. (b) The mass fraction of different chemical species in the composition from 12/22/2016 to 01/07/2017 and from 09/27/2017 to 10/01/2017.**

[Figure]

**Figure S6. (a) A scatterplot showing correlation of SMPS measurements to concurrent AMS measurements, and (b) correlation of DRX measurements to concurrent PDR measurements.**

[Figure]

**Figure S7. (a) Scatterplot of organic mass fraction and the PDR/AMS ratio. Points are color-coded by aerosol median diameter. (b) Scatterplot and correlation of aerosol median diameter with the DRX/AMS ratio. Points are color-coded by organic mass fraction. (c)**

**Scatterplot of organic mass fraction with the DRX/AMS ratio. Points are color-coded by aerosol median diameter. Points are sized by AMS mass concentration.**

In Fig. S7b the correlation between aerosol median diameter with the DRX/AMS ratio was lower than with PDR/AMS, with the most deviated points the result of smaller diameter particles. One possible reason for the low correlation coefficient with DRX is that particles were with smaller size than other periods, and the aerosol with diameters smaller than 100 nm would cause significant bias to DRX, due to its size detection limit. After excluding the points with smaller size, the $r^2$ increases to 0.52, which demonstrates the positive effect of aerosol size on optical instrument response.

[Figure]

**Figure S8. (a) Averaged mass size distribution of long-distance transported aerosol composition, and (b) averaged mass size distribution of fresh aerosol composition.**

[Figure]

[Figure]

(e) the map of fires and thermal anomalies from Terra and MODIS

**Fig. S9. GEOS-5 forecasts for the spatial distribution of sulfate aerosol AOT over North America (https://portal.nccs.nasa.gov) on 09/24/2017, 09/25/2017, 09/27/2017, and 09/28/2017 (above four), and the map of fires and thermal anomalies from Terra and MODIS (https://worldview.earthdata.nasa.gov) on 09/23/2017**

It is believed that the sulfate aerosol originated from the lower Mississippi Valley on 09/24/2017, as a result of multiple point-source fires clustered in the region. As time progresses, the sulfate aerosol distribution changed with synoptic downstream flow, and created the prominent band that passes over New York State on 09/28/2017. During this process, the aerosol was believed to have experienced long-distance transport.

[Figure]

**Fig. S10. Time series of estimated aerosol refractive index estimated using the PDR and DRX.**

[Figure]

**Figure S11. (a) The correlation scatterplots of aerosol refractive index and the PDR/AMS ratio. (b) The correlation scatterplot of aerosol refractive index and the DRX/AMS ratio. All points are color-coded by aerosol median diameter, and sized by AMS mass concentration**

**S3 Tables**

**Table S1. List of different dilution concentrations for each compound in the three test groups, and the volume median diameter of the generated size distribution.**

| Compound | Group 1 | Group 2 | Group 3 |
|---|---|---|---|
| NaNO$_3$   (g L$^{-1}$) | 0.75 | 3 | 6 |
| (NH$_4$)$_2$SO$_4$ (g L$^{-1}$) | 0.75 | 2.5 | 5 |
| Sucrose (g L$^{-1}$) | 0.45 | 1.5 | 3 |
| Adipic Acid (g L$^{-1}$) | 0.4 | 1.0 | 2.5 |
| Median Diameter (nm) | 153 | 202 | 231 |

**Table S2. Calculated relative Mie scattering flux of the four particle species ((NH$_4$)$_2$SO$_4$, NaNO$_3$, sucrose, adipic acid) analyzed by the PDR and DRX, and different dilution concentrations (#1 indicates Group 1; #2 indicates Group 2; #3 indicates Group 3).**

| Relative flux | (NH$_4$)$_2$SO$_4$ | NaNO$_3$ | sucrose | adipic acid |
|---|---|---|---|---|
| DRX(#1) | 3.37 | 2.56 | 3.34 | 2.50 |
| DRX(#2) | 4.72 | 5.04 | 5.04 | 3.19 |
| DRX(#3) | 5.41 | 5.54 | 5.54 | 3.72 |
| PDR(#1) | 0.79 | 0.59 | 0.86 | 0.69 |
| PDR(#2) | 1.31 | 0.93 | 1.45 | 0.96 |
| PDR(#3) | 1.59 | 1.03 | 1.67 | 1.24 |

**Table S3. Reference table RF for different refractive indices (TP refers to the time point for each hour of data, wavelength=880 nm, angle=90° )**

| Index/RF | TP1 | TP2 | TP3 | TP4 | TP5 | TP6 | TP7 | TP... |
|---|---|---|---|---|---|---|---|---|

| | | | | | | | | |
|------|------|------|------|------|------|------|------|-----|
| 1.2  | 0.40 | 0.39 | 0.39 | 0.40 | 0.40 | 0.41 | 0.41 | ... |
| 1.21 | 0.45 | 0.43 | 0.43 | 0.44 | 0.44 | 0.45 | 0.45 | ... |
| 1.22 | 0.49 | 0.47 | 0.47 | 0.48 | 0.49 | 0.49 | 0.50 | ... |
| 1.23 | 0.53 | 0.52 | 0.51 | 0.52 | 0.53 | 0.54 | 0.54 | ... |
| 1.24 | 0.58 | 0.56 | 0.56 | 0.57 | 0.58 | 0.59 | 0.59 | ... |
| 1.25 | 0.63 | 0.61 | 0.61 | 0.62 | 0.63 | 0.64 | 0.64 | ... |
| ...  | ...  | ...  | ...  | ...  | ...  | ...  | ...  | ... |
| 1.75 | 6.04 | 5.85 | 5.73 | 5.91 | 6.01 | 6.06 | 6.11 | ... |
| 1.76 | 6.22 | 6.02 | 5.91 | 6.09 | 6.19 | 6.25 | 6.30 | ... |
| 1.77 | 6.41 | 6.21 | 6.08 | 6.28 | 6.37 | 6.43 | 6.49 | ... |
| 1.78 | 6.60 | 6.39 | 6.26 | 6.46 | 6.56 | 6.62 | 6.69 | ... |
| 1.79 | 6.79 | 6.58 | 6.45 | 6.65 | 6.75 | 6.82 | 6.88 | ... |
| 1.8  | 6.99 | 6.77 | 6.63 | 6.84 | 6.95 | 7.02 | 7.08 | ... |